# Development of an Enzyme-Linked Immunosorbent Assay (ELISA) for the Quantification of ARID1A in Tissue Lysates

**DOI:** 10.3390/cancers15164096

**Published:** 2023-08-14

**Authors:** Manuel Hinsberger, Julia Becker-Kettern, Wiebke M. Jürgens-Wemheuer, Joachim Oertel, Walter J. Schulz-Schaeffer

**Affiliations:** 1Institute for Neuropathology, Medical Faculty, Saarland University, Building 90.3, 66421 Homburg, Saar, Germanyjulia.kettern@uks.eu (J.B.-K.); wiebke.wemheuer@uks.eu (W.M.J.-W.); 2Department of Neurosurgery, Medical Faculty, Saarland University, Building 90.3, 66421 Homburg, Saar, Germany; joachim.oertel@uks.eu

**Keywords:** ARID1A, ELISA, western blot, SWI/SNF complex, BAF complex, gynecologic tumors

## Abstract

**Simple Summary:**

*ARID1A* is frequently mutated in cancers, especially in gynecologic ones, but very little attention has been paid to determine ARID1A levels in tumor specimen. Therefore, the main aim of this work was to develop an enzyme-linked immunosorbent assay (ELISA) to measure ARID1A protein levels in an accurate and precise manner. In addition, we performed western blot analysis to validate the ELISA results. As we can show the high specificity, accuracy, and precision of our assay, further research can now focus on whether ARID1A protein levels are of predictive value, as most current methods analyze its gene *ARID1A* only. In terms of functionality, ARID1A is a member of the SWI/SNF complex, but it is not clear how mutations in members of this chromatin remodeling complex lead to tumorigenesis. We hope to facilitate the quantitative analysis of proteins involved in tumorigenesis with this detailed ELISA protocol for the nuclear protein ARID1A.

**Abstract:**

ARID1A is a subunit of the mammalian SWI/SNF complex, which is thought to regulate gene expression through restructuring chromatin structures. Its gene *ARID1A* is frequently mutated and ARID1A levels are lowered in several human cancers, especially gynecologic ones. A functional ARID1A loss may have prognostic or predictive value in terms of therapeutic strategies but has not been proposed based on a quantitative method. Hardly any literature is available on ARID1A levels in tumor samples. We developed an indirect enzyme-linked immunosorbent assay (ELISA) for ARID1A based on the current EMA and FDA criteria. We demonstrated that our ELISA provides the objective, accurate, and precise quantification of ARID1A concentrations in recombinant protein solutions, cell culture standards, and tissue lysates of tumors. A standard curve analysis yielded a ‘goodness of fit’ of R^2^ = 0.99. Standards measured on several plates and days achieved an inter-assay accuracy of 90.26% and an inter-assay precision with a coefficient of variation of 4.53%. When tumor lysates were prepared and measured multiple times, our method had an inter-assay precision with a coefficient of variation of 11.78%. We believe that our suggested method ensures a high reproducibility and can be used for a high sample throughput to determine the ARID1A concentration in different tumor entities. The application of our ELISA on various tumor and control tissues will allow us to explore whether quantitative ARID1A measurements in tumor samples are of predictive value.

## 1. Introduction

As the mammalian homolog of the SWI/SNF complex (switch/sucrose non-fermentable complex), the BAF complex (short for cell BRG1/BRM-associated factor complex) has its role as an ATP-dependent chromatin remodeling complex in the nucleus and contains up to 15 subunits with varying combinations. These are encoded by 29 genes and take three differently assembled forms in humans: canonical, polybromo-associated, and non-canonical BAF (BAF, PBAF, and ncBAF, respectively) [1], all of which share a characteristic ATPase module, which is highly conserved and functionally important [2]. The AT-rich interaction domain 1A (ARID1A), which has several aliases such as BAF250a and SMARCF1, is exclusively contained in forms of the BAF complex. In general, chromatin-remodeling complexes maintain three distinct functions according to Clapier et al. (2017): nucleosome assembly, chromatin access, and nucleosome editing [2]. Furthermore, it was shown that the function of a single mammalian SWI/SNF complex correlates with the unique form. As the AT-rich interacting domain (ARID) in ARID1A has its own role in bringing chromatin-remodeling complexes to DNA sequences [3], Raab et al. (2015) propose a model where the function of a complex is related to one of the three ARID proteins, ARID1A, ARID1B, or ARID2, as they recruit their own clusters of transcription factors and have co-operative as well as competitive interactions [4].

In 2013, two independent studies described that approximately 20% of all human cancer types harbor mutations in the mammalian SWI/SNF complex and that the *ARID1A* gene has the highest mutational rate between 4.6% and 9% [5,6]. Since *ARID1A* mutations were described first in gynecologic tumors, it seems as if their role in both endometrioid and clear-cell ovarian cancers is of particular importance [7,8]. Kadoch et al. (2013) found an *ARID1A* mutation frequency rate of nearly 45.2% in those tumor entities [6]. An in-depth analysis highlighted that a G2087 mutation to valine, arginine, or glutamic acid and the truncating Y2254* mutation are the most frequent missense mutations of *ARID1A* [1]. These mutations seem to lead to a partial or complete loss of function [9]. Another cause for the loss of ARID1A can be an *ARID1A* promoter hypermethylation and it is likely that both mechanisms may work in combination for a tumor to achieve a complete loss of ARID1A proteins [10,11]. Due to its central role in transcriptional regulation and the maintenance of DNA integrity, ARID1A harbors the potential to be a powerful tumor suppressor [5,6].

Consequently, the quantification of ARID1A seems to be of great importance. On the one hand, preserved ARID1A is regarded as a prognostic marker in some tumors, e.g., for stage I/II ovarian clear cell carcinoma and endometrioid ovarian cancers [12]. As a predictive marker in estrogen-receptor-positive (ER+) breast cancer, *ARID1A*-inactivating mutations lead to a change of the cell type and convey resistance against estrogen-receptor degrading treatment [13]. On the other hand, ARID1A deficiencies may open ways in the future to promote the cell death of tumor cells, as ARID1A loss impairs genome stability in the tumor cells with their already high mutational burden, by several mechanisms: base excision repair as well as mismatch repair are often impaired, and the formation of R-loops and DNA catenation occur more frequently. Targeting complementary pathways that will try to compensate for ARID1A failure is a possibility to provide a synthetic lethal effect in those tumor cells [11]. Several clinical trials are currently investigating the effects of inhibiting ARID1A-modulated signaling pathways; e.g., Nivolumab, a PD-1 (programmed cell death protein 1) inhibitor, is being tested in a phase II trial NCT04957615 [14] and the outcome of treatment with tazemetostat, an EZH2 (enhancer of zeste homolog 2) inhibitor, in solid tumors harboring an *ARID1A* mutation is being investigated in a phase II study NCT05023655 [15].

On the technical side, there is a shortage of quantitative methods measuring the concentration of ARID1A in an easy and accurate way. Few commercial ELISA kits are available, which are, on the one hand, expensive (300 EUR per 96-well plate and higher) and, on the other hand, have not been used or referred to for research (see Appendix A). Also, the precision and accuracy of those ELISAs have been determined with limited sample numbers, if stated at all. Additionally, in the field of (neuro)pathology, immunohistochemistry (IHC) is often the method of choice, yet it has its limitations when visually estimating different protein expression levels. IHC remains a semiquantitative method and there are no calibrators on each slide to ensure that differences in fixation, sectioning, antigen retrieval, etc. have an impact on the interpretation of the signal intensity [16,17]. Furthermore, it is known that there is only a linear relationship between the amount of antigen and staining intensity at rather low levels as the assay is saturated at high levels [16,18]. In addition to an inter-assay inconsistency, there is also an intra-observer variability whose extent was shown to be up to 50% of ineffectively ranked signal intensities if one pathologist values the same tumor twice [19]. Given these two major problems in IHC, we aimed to develop an enzyme-linked immunosorbent assay (ELISA) to quantify ARID1A protein levels in a more accurate and objective manner [20,21,22]. For this purpose, we chose to use an indirect ELISA, in which a cell lysate coats the wells. We decided to fractionate the tissue lysate into four subcellular compartments (cytosol, membrane, free nucleus, and chromatin) to achieve adequately extracted ARID1A from the nucleus and have the other cellular fractions serving as internal controls for the assay. Our goal was to establish and validate an ELISA with high precision and accuracy in the quantification of ARID1A levels, following the current EMA and FDA guidelines for ligand-binding assays [23,24].

## 2. Materials and Methods

### 2.1. Materials

To provide optimal binding conditions for the ARID1A protein to the plates, we analyzed the hydrophobicity of ARID1A using GRAVY (grand average of hydropathicity index), as proposed by Kyte and Doolittle (1982) [25], by Expasy’s ProtParam Server [26]. Accordingly, ARID1A (GRAVY score = −0.78) was classified as hydrophilic [27], and we chose a NUNC MaxiSorp plate (750-0083, VWR, Darmstadt, Germany).

As a quantitative standard, we used an ARID1A fragment (ABN-H00008289-Q01, Abnova, Taipei, Taiwan) containing amino acids 1216 to 1325 (NCBI Reference Sequence: NP_006006.3) and, to ensure specificity, we used an *ARID1A* knockout HEK-293T cell pellet (ab278824, Abcam, Cambridge, UK) and the corresponding control cell pellet. Western blot analysis was performed using *ARID1A* wild-type and knockout western blot lysates (ab257250, Abcam, UK). The primary antibody ab182560 (Abcam, UK) binds a region between amino acids 1200 to 1350, covering the fragment’s epitope sufficiently. Table 1 provides the details for all solutions used in this assay.

We performed tumor tissue analysis with 37 meningiomas resected at the Department of Neurosurgery, Saarland University, between 2018 and 2021, and subsequently diagnosed after neuropathological expertise according to the WHO classification of the tumors of the central nervous system [28]. Ethical approval was given by the ethics committee of the Saarland medical council (no. 51/22).

### 2.2. Lysate Preparation

We chose the ‘Subcellular Protein Fractionation Kit for Tissues’ (87790, Thermo Fisher Scientific, USA) as this kit contains strainers to retain excess connective tissue or bone fragments that could contaminate the lysate and, thus, impair correct measurement. This characteristic was indispensable because tumor tissue is often attached to ligaments or other connective tissue, and we aimed at minimizing the variation in tissue quality due to inadequate lysate preparation for downstream applications. In brief, 100 mg tumor tissue were weighed and placed into the Precellys Ceramic tubes (10144-496, VWR, Germany), and 1000 µL of cytoplasmatic extraction buffer (CEB) for tissues was added. The sample was homogenized two times for 20 s on speed 2 within the Minilys homogenizator (432-0274, VWR, Germany). The lysate was placed onto the cell strainer within a falcon tube and centrifuged at 500× *g* for 5 min. The treated tissue lysate was then fractionated into cytoplasmic, membrane-bound, free nucleus, and chromatin-bound fractions according to manufacturer’s instructions. This resulted in approximately 1000, 650, 225, and 170 µL of cytoplasmic, membrane-bound, free nucleus, and chromatin-bound fraction, respectively, with a total protein concentration ranging in average between 500 µg/mL and 10 mg/mL protein with the highest values seen in the cytosol. Knockout and wild-type cell pellets were prepared using 20 µL packed cell volume and following the manufacturer’s instructions corresponding to the protocol ‘Subcellular Protein Fractionation Kit for Cultured Cells’ (78840, Thermo Fisher Scientific, USA).

### 2.3. Determination of Protein Concentration

To measure protein concentrations in the lysates to ensure plate binding saturation, we used a ‘Pierce bicinchoninic acid (BCA) protein assay kit’ (23225, Thermo Fisher Scientific, USA) as the manufacturer’s instructions allow the use of our buffers from the Subcellular Protein Fractionation Kit, including cytoplasmatic extraction buffer (CEB), membrane extraction buffer (MEB), and nuclear extraction buffer (NEB). In brief, standards reaching from 20 µg/mL to 2000 µg/mL as well as 1:26 diluted lysates (total volume 25 µL) were added on the NUNC MaxiSorp plate. The BCA working reagent was prepared, and 200 µL were pipetted per well. Incubation started with gentle shaking on a plate shaker for 30 s, then the plate was covered with parafilm and incubated at 37 °C for 30 min. Results were analyzed using the FLUOstar Omega Microplate Reader (BMG LABTECH, Ortenberg, Germany) at 562 nm.

### 2.4. Statistical Analysis

Curve analysis and non-parametric fits as well as Student’s *t*-tests were performed using GraphPad Prism (version 9.4.1). Interpolation of data should be performed by the simplest fit available; therefore, a 4-parameter logistic fit (4PL) has been used. Some measurements needed a 5-parameter logistic fit (5PL) to improve the curve’s fit.

### 2.5. ELISA Protocol

To create a reliable and reproducible protocol, we first optimized each step, and then standardized the protocol. Blocking and washing steps proved to be critical, and several buffer solutions had to be tested. Final optimized solution recipes are given in Table 1. Table 2 is a comprehensive overview of our indirect ELISA protocol for ARID1A. In brief, the wells of the NUNC MaxiSorp plates were coated overnight at 4 °C without agitation and 100 µL per well of the cell fractions at a dilution of 1:26. During each incubation period, the plates were covered. After washing three times with 200 µL Phosphate-buffered saline supplemented with 0.1% Tween-20 (PBS-T 0.1%) for five minutes, blocking was performed with 200 µL PBS-T (0.05%) with 0.2% (*w*/*v*) casein at room temperature (=RT) for 2 h on a plate shaker. Then, the primary antibody was diluted 1:1000 in antibody solution, and 100 µL were added to the wells and then incubated at RT for 2 h. The plate was treated with the secondary antibody diluted 1:2 analogously (compared in Table 2). After another washing step, 5 mg of OPD (o-Phenylenediamine dihydrochloride) was diluted in 9 mL of distilled water and 1 mL of peroxide stable substrate. Then, 100 µL of this substrate solution was applied and incubated for 30 min in the dark. Addition of 100 µL of 2.5 M H_2_SO_4_ increased the measurement signal without enhancing the background. Therefore, absorbance measurements were performed at 490 nm using the FLUOstar Omega Microplate Reader.

### 2.6. Standard Curve Analysis and Acceptance Criteria

For data analysis, we analyzed the absorbances with the MARS software provided by the FLUOstar Omega Microplate Reader, applying background corrections and using several replicates of eight differently concentrated calibration standards. Additionally, we measured six incurred samples from tumor tissue as quality controls at least in duplicate. Standard curve analysis was performed using GraphPad Prism (version 9.4.1 for Windows, GraphPad Software, La Jolla, CA, USA) using a 5-parameter logistic with weighted fit (1/Y^2^) as this was the most efficient setting (sometimes including anchor points at the high or low end). To maintain high reproducibility of the assay, run acceptance criteria included accuracy > 80% of the actual value (>75% for lower limit of quantification, LLOQ, and upper limit of quantification, ULOQ) and precision <20% (expressed as coefficient of variation (CV), <25% for LLOQ and ULOQ) in a minimum of 75% of the standards. Additionally, correctness (R^2^) should be above 0.97 and, after excluding standards failing the above-described criteria, at least six standards should remain. For incurred sample reanalysis (ISR), our aim is that 67% of the samples should be within ±30% of the mean [23,24].

Accuracy and precision are essential values when evaluating the assay’s performance; therefore, a short definition is given in the following. Since accuracy is expressed as a percent of the nominal value, it defines the degree of closeness of the measured value to the nominal or known true value. Furthermore, precision is expressed as the mean of each analyte’s CV (coefficient of variation): the standard deviation divided by the mean of an analyte, multiplied with 100 to express the results in percentage [23,24].

### 2.7. Western Blot

Western blot was performed according to the method of Laemmli (1970) [29]. Equal amounts of protein from cell lysate or tissue homogenate (30 µg) were resolved electrophoretically using anykD-SDS-PAGE gels in a TRIS-glycine buffer system (4569033, Bio-Rad Laboratories, Hercules, CA, USA), and then transferred to a 0.2 µm nitrocellulose membrane (1620112, Bio-Rad Laboratories, USA) for either 30 min at 100 V when analyzing the ARID1A fragment (37.84 kDa) or 60 min at 100 V when analyzing full-length ARID1A (242 kDa). Then, membranes were incubated with blocking solution (see Table 1) overnight at 4 °C. After washing the membrane with TBS (10 mM TrisHCl, 100 mM NaCl, pH 7,8) added with 0.05% Tween-20 (TBS-T), incubation with the primary antibodies against ARID1A (ab182560, Abcam, UK) diluted 1:2000 in TBS-T or alpha-Tubulin (ab7291, Abcam, UK) diluted 1:2200 in TBS-T as loading control was performed for 1.5 h at RT. After thoroughly washing the membranes with TBS-T, they were incubated for 1.5 h at RT with HRP-conjugated anti-rabbit secondary antibody (K4003, Dako, Denmark, diluted 1:500 in TBS-T) or anti-mouse antibody (K400111, Dako, Denmark, diluted 1:500 in TBS-T) for ARID1A or alpha-Tubulin, respectively. After repeated washing, signals were visualized using the Amersham ECL Select kit (RPN2235, VWR, Germany) according to manufacturer’s instructions and the membranes were then exposed to Amersham ECL Hyperfilms (28-9068-37, VWR, Germany). Finally, films were incubated with developing reagent (65282194, Heraeus Kulzer, Wehrheim, Germany) and fixing reagent (65282259, Heraeus Kulzer, Germany). For analysis, each membrane included at least one lane with a molecular weight marker (161-0394, Bio-Rad Laboratories, USA).

## 3. Results

To validate the provided technically simple and robust enzyme-linked immunosorbent assay (ELISA) protocol, we first analyzed our calibration curves and determined a quantitative range between 0.78 ng/mL and 50 ng/mL ARID1A. Because this range allows optimal ARID1A analysis in our tumor lysate dilutions, we consequently performed specificity analyses in *ARID1A* wild-type and knockout cell lysates, which revealed a significant decrease in ARID1A concentration in the three subcellular fractions, cytosol, free nucleus, and chromatin. Moreover, all these results were confirmed by a complementary western blot analysis.

### 3.1. Calibration Curve

The sensitivity (or detection limit, DL) of our ELISA was 0.39 ng/mL as this was the minimal concentration that could be detected with acceptable precision and accuracy (see Figure 1A). However, when using all values for a standard curve of all experiments (=AE model), this standard at 0.39 ng/mL failed the acceptance criteria of both precision and accuracy, so the quantitative range was set from 0.78 ng/mL (lower limit of quantification, LLOQ) to 50 ng/mL (upper limit of quantification, ULOQ).

The coefficient of determination (R^2^), which represents the ‘goodness of fit‘, regarding the AE standard curve was 0.987 in a very robust model that included all experiments conducted for this paper. When considering each plate individually (=EP model), nine out of eleven tests for the resulting calibration curves had an R^2^ value above 0.99. The theoretical model, thus, describes reliably the dependence of absorbance signals and ARID1A standard concentrations.

With this protocol, we were able to measure ARID1A concentrations in the four different subcellular compartments. The ARID1A levels were ranked as follows, in descending order: free nucleus, chromatin, cytosol, and membrane (see Figure 1A).

To put our results in relation to other established but less quantitative methods, we performed western blot analyses. We analyzed a tumor lysate prepared according to the above-described protocol (see Appendix A). Concordantly, presumable ARID1A signals in the free nucleus and chromatin fractions were higher than those in the cytosol and membrane (Appendix A). When using the ARID1A fragment in several dilutions (0.5–1000 ng/mL, Figure 1B), we observed bands at the expected height (37.84 kDa) with adequate signal intensity reduction from higher to lower concentrations. A densitometric analysis shows a sigmoidal shape of the band intensity as a function of the fragment concentration, consistent with our results in the ELISA (see Appendix A). Additional bands were seen at a lower weight down to slightly less than 20 kDa, especially in the higher concentrations.

### 3.2. Precision and Accuracy: Standards and Incurred Sample Reanalysis

In the AE model, we detected an inter-assay accuracy of 90.26% and an inter-assay precision with a coefficient of variation of 4.53%. In contrast, the EP model showed an inter-assay accuracy of 93.16% and an inter-assay precision with a coefficient of variation of 4.45% (see Table 3). In addition, our assay achieved an intra-assay precision of 4.05% (see Appendix A).

Besides recombinant standards, we repeatedly measured tumor tissue samples. We re-measured previously prepared lysates (see Table 3, Column 4) and compared them with new lysates obtained from the same tumors but different regions (see Table 3, Column 5). Membrane fractions were measured, too, but many of them produced signals below the LLOQ, resulting in results that are not quantifiable. We observed an inter-assay precision (in this scenario, the deviation of a single value compared to the mean of repeated measurements) with a coefficient of variation of 11.78% based on all fractions, excluding the membrane fraction.

### 3.3. Specificity

To evaluate the specificity of the primary antibody ab182560 used for our ARID1A-ELISA, we used two cell pellets derived from HEK-293T cells with one bearing an *ARID1A* knockout (=KO) and the other one being *ARID1A* wild-type (=WT). In theory, a complete knockout, as caused here by a 11 bp homozygous deletion in exon 1, leading to a truncated ARID1A protein, should deplete the ARID1A signal in our ELISA measurement. In the knockout cell line, a signal decrease across all four fractions is visible (see Figure 2A). *T*-tests showed a significant difference in cytosol, free nucleus, and chromatin. While membrane fractions failed to show a significant difference, a decrease of 71% is still visible. However, signals in the membrane fraction of the knockout were, in general, more than six-fold lower than in the lowest fraction of the wild type.

To confirm the obtained ELISA results, we performed a western blot analysis of analogous lysates suited for SDS-PAGE (ab257250). A band at nearly 242 kDa is only visible in the wild-type lysate (see Figure 2B and Appendix A). Performing a densitometry analysis of each lane using ImageJ [30], a Student’s *t*-test revealed a highly significant signal intensity reduction (*p* ≤ 0.0001) in knockout cells (see Figure 2D). We next asked if the subcellular fractionation of cell lysates from *ARID1A* wild-type and knockout (ab278824) according to Section 2.2 reveals similar results in western blot and ELISA: a significant decrease (*p* ≤ 0.01) in signal intensity at the 242 kDa band is evident in all four fractions (Figure 2C,E).

We see a band at approximately 60 kDa (Figure 2C), which is also present in both lanes in ab257250 (see Appendix A). As it can be seen to be independent of the *ARID1A* deletion, it might be unspecific binding. In addition, only in the chromatin fractions is an additional band of almost 16 kDa present (Appendix A)

In the enzyme-linked immunosorbent assays, chromatin fractions had high background signals at about 40% of the back-calculated concentration and varied only minimally (compare Figure 2A). Detailed analysis showed that our secondary antibody has unspecific binding in this fraction: at the beginning of our experiments, we worked with an undiluted secondary antibody and had a significant decrease of 11% in *ARID1A* KO. A higher dilution showed that 1:2 is the best dilution, as it enables high sensitivity and acceptable specificity, including a more significant decrease of 58% in *ARID1A* knockout cells. Changing the secondary antibody towards ab6721 (1:120.000, Abcam, UK) demonstrated no improvement for our assay.

To identify the cause for this background signal, we performed experiments coating pure buffer solutions from our lysate fractionation (CEB, MEB, NEB, and NEB supplemented with MNase; see Section 2.2) and analyzed whether primary and/or secondary antibodies produced a signal. We noticed that incubation with both antibodies produced low but measurable signals in the chromatin fraction, but not in any other (see Appendix A). When incubating the plate only with the secondary antibody K4003 diluted 1:2 after blocking, there was a significantly (***, *p* = 0.0009) higher signal in the chromatin fraction but in no other, indicating the unspecific interaction of our secondary antibody with the buffer ingredients (see Appendix A). Since the MNase, which has a molecular weight of 16.8 kDa, is added to the chromatin fraction but not to the free nucleus fraction and both contain the nuclear extraction buffer (NEB), we assume that this protein interferes with our assay. This is further strengthened by the 16 kDa band in the western blot analysis of the four lysates and the *ARID1A* wild-type and knockout lysates, as it is only visible in the chromatin fractions (Appendix A).

### 3.4. Spike Recovery

To evaluate the spike recovery in our assay, we added three different amounts of the ARID1A fragment ranging from 6.25 to 25 ng/mL to the prepared subcellular fractions of the *ARID1A* knockout cell pellet and subtracted the back-calculated ARID1A concentrations of the corresponding unspiked lysates. The spike recovery was then expressed as the percentage of the resulting ARID1A concentration divided by the nominal concentration. This resulted in a mean of 18%, 0%, 16%, and 6% recovery in our four fractions cytosol, membrane, free nucleus, and chromatin (see Figure 3A). In addition, we analyzed the *ARID1A* wild-type cell pellet, as well as two tumor tissues in ELISA, to confirm the above-described results, and found similarly low recovery rates, suggesting neither methodical nor technical errors.

To understand these surprising results, we tested the fragment stability as we thought this to be the potentially limiting step for the spike recovery analysis. Therefore, we prepared several standard dilutions, incubated them for 0, 45 min, and 90 min on ice, and determined the reduction of the absorbance signals over time (Figure 3B). Indeed, we saw a massive loss of signal in all standard concentrations over the time (Figure 3B).

Next, to validate the degradation processes, we performed the western blot analysis with our fragment (see Appendix A). Equal amounts of the ARID1A fragment were incubated at 4 °C for up to 24 h. Initially, there was only a single band at 37 kDa, which matches with the predicted fragment length. Over time, another band at almost 18 kDa appeared, and its intensity increased with the incubation time, indicating degradation (Appendix A).

Moreover, to explain why membrane fractions show zero recovery, we spiked known concentrations of the ARID1A fragment in several buffers that were properly diluted 1:26 as described in Section 2.5, but contained no tissue. We saw a complete recovery in the cytosol extraction buffer (93%), nuclear extraction buffer (98%), and nuclear extraction buffer spiked with MNase (62%), but a 0% recovery rate in the membrane extraction buffer (see Appendix A).

### 3.5. Minimal Required Dilution (MRD)

To determine to what extent tissue lysates could be diluted without affecting the measured ARID1A concentrations due to a saturation of the binding surface, we performed serial dilutions of the lysates and converted the measured absorption values into ARID1A concentrations using the recombinant ARID1A as a standard with known concentrations and quantified total protein using an BCA assay (Figure 4A). We observed an exponential decrease in signal when the total protein concentration was below 20.08 µg/mL (Figure 4A). In contrast, there was no increase in absorption when the lysates had a concentration higher than 20.08 µg/mL of total protein. The latter shows that no ‘hook effect’ is visible, which would mean suppression by matrix effects. Therefore, we assumed that a minimum of about 20 µg/mL is necessary to ensure binding surface saturation (=minimum required dilution, MRD). This value is reflected by general recommendations to minimally coat the plate with a total of 20 µg/mL of protein. Additionally, the BCA assay fails if there are total protein concentrations lower than 20 µg/mL as this is the lower limit of quantification in this assay.

### 3.6. Parallelism

To evaluate the parallelism between standards and tumor lysates, serial dilutions of both were coated on the plate and, after measurement, the absorptions were plotted against the dilution with a corresponding dilution factor of 2 (Figure 4B). As both curves show high parallelism, we could show that there is no matrix effect caused, for example., by protein-protein interaction or precipitation in the tumor lysates compared to the recombinant protein (Figure 4B, left). Only if both curves show similar slopes is this assumption valid [31]. Thus, the absorption signals in lysates can be interpolated by the standard curve into estimated protein concentrations. As a point of interest, we saw that ARID1A absorption signals do not decrease at lower dilutions, and we think this is due to the saturation of binding surfaces (Figure 4B, right) and has no relevance in interpreting parallelism. It underlines that ARID1A levels above 20 µg/mL are stable, as we see no hook effect, but can be interpolated by the standard curve as well.

## 4. Discussion

Since ARID1A has taken a central role in tumorigenesis and in many different tumor entities over the last years [1,5,7,8], it is of importance that quantitative methods are optimized and validated before making their way into clinical application. To do so, we established an indirect ELISA (enzyme-linked immunosorbent assay) that measures the actual ARID1A protein concentrations in tumor lysates with extraordinarily high precision and accuracy (compare Appendix A). The superiority of the EP model in comparison to the AE model in accuracy (see Table 3) may be explained by a less varying standard curve as little differences in, for example, temperature, buffer quality, or air humidity were better represented. Concerning tumor tissue lysates, our analysis demonstrates, on the one hand, that it is necessary to prepare lysates in a standardized protocol as inconsistent tissue weighing, buffer preparations, or incubation periods lead to higher variations in the results (compare Table 3, Column 4). On the other hand, our results show that there is an increase in deviation from ‘same lysates’ to ‘different lysates’ (Table 3, Columns 4 and 5), which might be due to a different ARID1A expression pattern in different tumor regions and is uninfluenceable. Since EMA and FDA allow a maximum deviation of 30% and we remained well under this limit, this does not affect the quality of our assay [23,24]. When analyzing the total ARID1A levels in cell cultures or tissue homogenates, it might be advantageous to sum the four fractions—cytosol, membrane, free nucleus, and chromatin—and perform a theoretical total ARID1A protein measurement to compensate for carryover during subcellular fractionation. This total ARID1A may be useful for exploring threshold values in tumors with *ARID1A* dysfunction. Nevertheless, fractionation has its benefits as it ensures the high extraction rate of nuclear and DNA-bound ARID1A, enhances the measurement through enriching ARID1A, and allows a new biochemical analysis of ARID1A.

In this work, we demonstrated that our ELISA is specific for ARID1A (see Figure 2A), but chromatin fractions showed a relatively high background. However, we were able to show that these cross-reactions were related to the secondary antibodies. Both western blot and ELISA analyses suggest that the 16.8 kDa heavy MNase is responsible for this background (Appendix A), which is added to the NEB buffer before preparing the chromatin fraction. This assumption is in line with previously reported cross-reactivities for human monoclonal antibodies [32]. Since chromatin is essential for ARID1A analysis, we aimed to improve a new model for this fraction. We tested 12 tumor chromatin fractions and asked whether the background signal could be reliably quantified (Appendix A). In fact, there is a background signal that has almost equal absorbance values expressed as optical density (OD), with only two outliers among the twelve tumors analyzed. The mean OD is 0.0315 (95% CI: 0.0208–0.04216), which is the value obtained in chromatin fractions of the twelve tumors when the OD signal was generated only by the secondary antibody K4003, reflecting the nonspecific binding in tumor tissue matrix. We then re-analyzed the *ARID1A* knockout (KO) and wild-type (WT) lysates and found a complete annihilation of the knockout signal (0.027 OD) in the chromatin fraction when the blank value of 0.0315 is subtracted (see Appendix A). Wild-type signals in the ‘adjusted model’ were barely affected, as ARID1A levels decreased by only 22.2%, which is not significant compared to values in the ’normal model’ (*p* > 0.05). This could be a great improvement for further ARID1A analysis in cell cultures or tissue homogenates, as it is very straightforward to implement. However, these results should be further validated.

Because of the large variability of genetic alterations in tumors, potential difficulties that could affect the correct interpretation of measurement results include interactions with *ARID1A* mutants, as many tumors carry *ARID1A* point mutations [5,6]. The most frequent missense mutation of *ARID1A*, however, leads to an exchange of the amino acid glycine at position 2087 to valine, arginine, or glutamic acid (G2087V, G2087R, G2087E; [1]). Another frequent missense mutation Y2254* results in a 31 amino acid deletion of the ARID1A C-terminus. Mashtalir et al. (2018) found that the G2087R mutation can result in lower ARID1A expression in HEK-293T cells via increased proteasome-mediated degradation, whereas the truncating mutation Y2254* leads to a functional loss of ARID1A [1]. The exchange at position 2078 should not interfere with our antibody, but Y2254* could produce false-positive signals that do not correlate with a functional protein. Care should therefore be taken when extrapolating ELISA results to tumor samples for diagnostic purposes. In contrast, the more frequent mutation G2087R of *ARID1A* should be detectable with our antibody with regard to the expression loss and, thus, be helpful for diagnostic purposes, reflected by an ARID1A loss.

Also, we used an ARID1A fragment that worked very well with our antibody, but it appears obvious that a fragment cannot fully represent a full-length protein, and, therefore, all interpolations must take this aspect into account. During our experiments, we showed that tumor lysates worked very well with our protocol and specificity, as well as quantitative range, ensuring a high quality. In some assays, fragments still harbor improvements in the affinity between antigen and antibody, so the use of an ARID1A fragment may be useful in order to obtain absolute ARID1A values [33]. For example, Grebenchtchikov et al. (2004) produced protein fragments to reduce the antibody’s cross-reactivity [33]. In our assay, there was no full-length protein available, but a fragment of ARID1A (aa. 1216–1325) worked very well with our antibody ab182560 (immunogenic amino acids 1200–1350). While the fragment is very useful for generating a standard curve (see Figure 1A,B), it is not useful for spike recovery analysis (Figure 3A). We were able to detect the degradation of the fragment in both western blot and ELISA, suggesting that signal loss occurs even in the absence of tissue contact, resulting in lower recovery rates (0–18%). Protease activity may still be present even after protease inhibition [34], leading to fragment degradation, making the interpretation of the results more ambiguous. The development of the full-length ARID1A, for example, in *E. coli*, may optimize this ELISA protocol in the future, formally leading to a complete validation. For this assay, we argue that incurred sample reanalysis and parallelism, both performed in tissue matrix, provide a higher level of interpretability. Since both demonstrate excellent results (see Table 3 and Figure 4B), we see no limitations for our assay. In addition, it is important to note that the degradation of our ARID1A fragment during the preparation of standard dilutions is extremely unlikely because of the short time interval between the preparation of multiple tubes and pipetting them onto the plate. This can also be seen from the fact that 11 summarized standard curves in the AE model are subject to a high precision as well as accuracy (Figure 1A, Table 3), which could not be achieved as soon as the protein degradation would distort the signal values of a single assay. Therefore, a coefficient of determination (R^2^) of 0.99 is an indication of the high representability of the standard curve that can be obtained with the fragment (see Section 3.1).

A comprehensive analysis of the ARID1A distribution over the four fractions in lysates shows unexpected distributions (see Figure 2A). One could think that the highest amounts of ARID1A—part of the chromatin remodeling complex and thought to play a major role in connecting BAF complexes to DNA regions [4]—would be present in the chromatin fraction, but it is obvious that 60.7% of the signal comes from the free nucleus when considering the ‘adjusted model’ described above. Overall, the cytosol harbors 23.3%, the membrane 2.5%, and the chromatin 13.5% of the intracellular ARID1A. As ARID1A was reported to be a nucleocytoplasmic protein [35], we can confirm this assumption as signals in both locations were clearly visible (compare Figure 2A). The membrane fractions have low ARID1A levels, which may be a false-negative conclusion, because we have evidence of the low recovery of the ARID1A fragment in the membrane extraction buffer (MEB), which is not the case with other buffers (see Appendix A). Additionally, this is most likely the cause for the observed failing significance between the *ARID1A* KO and WT lysates here (Figure 2A). Further explanations for the lower levels in chromatin compared to free nucleus fractions could be, on the one hand, that the BAF complexes harboring ARID1A and being bound to DNA were washed out in the preceding free nucleus fraction and, therefore, produced non-soluble—misclassified—ARID1A concentrations in this fraction. To explore this question, we performed ELISA experiments with analogously prepared samples and a Histon H3 antibody (ab1791, 1:1000, Abcam, UK) that has been already used in ELISA [36], which should predominantly be detectable in the chromatin fraction. As expected, the highest Histon H3 amounts were indeed found in the chromatin fraction. However, a measurable signal in the free nucleus fractions supports that normally chromatin-bound protein can, under the extraction conditions used, be found in the free nucleus fraction (Appendix A). Currently, we cannot fully answer this question, and further research could focus on this issue. On the other hand, it is possible that soluble ARID1A in the free nucleus fraction is of functional relevance and depicts a homeostatic state of both the ARID1A and BAF complexes.

Last but not least, the AT-rich interaction domain 1B (ARID1B) is a protein that shares a nearly 60% sequence identity with ARID1A [37]. In terms of functionality, the two homologs ARID1A and ARID1B are mutually exclusive when it comes to the assembly of the BAF complex [37]. Those canonical BAF complexes consist of either ARID1A or ARID1B [1,37], which determines the consecutive changes in DNA-binding behavior [4]. On a functional level, it is possible that ARID1B replaces ARID1A at least to some extent [38]. It is, therefore, of importance to solely measure ARID1A. Some previous work has discussed the notion that modern antibodies bind protein sequences that are highly probable for ARID1A and, therefore, allow discrimination between these two paralogs [37]. When inspecting the amino acid sequence targeted by the antibody (aa. 1200–1350), we found that ARID1A and ARID1B share 43.7% different, 39.1% biochemically identical, 8.6% strongly similar, and 8.6% weakly similar amino acids analyzed by the Multiple Sequence Alignment tool ‘Clustal Omega’ from EMBL-EBI [39]. The antibody’s binding epitope contains a 47 amino acids-long sequence (aa. 1289–1335), where ARID1A and ARID1B share no identical amino acid, and, therefore, one could argue that an interaction in the preceding binding region (aa. 1200–1288) is still possible. Since we used a very sensitive method, which wants to exclusively measure ARID1A, we considered this interaction, but we did not detect any interference in our ELISA in the membrane, cytosol, and free nucleus fractions. When comparing the *ARID1A* wild-type and knockout cell lines, both fractions—cytosol as well as free nucleus—show a significant decrease in knockout cells with a more than 90% reduction in back-calculated ARID1A concentrations (see Figure 2A). These results can be confirmed with western blot (WB) either in ab257250, an *ARID1A* WT and KO lysate suited for WB (see Figure 2B,D), or ab278824, an *ARID1A* WT and KO cell pellet fractionated in four subcellular compartments (see Figure 2C,E). Since ARID1B should still be present in both compartments [38], we therefore argue that the antibody can be termed ‘specific’. When comparing the antibody’s binding epitope with the protein sequence of the AT-rich interaction domain 2 (ARID2) analogously in the Clustal Omega tool, the results were as follows: ARID1A and ARID2 share 53.6% different, 15.2% biochemically identical, 18.5% strongly similar, and 12.6% weakly similar amino acids in ARID1A’s sequence 1200–1350 [39]. Moreover, ARID2 has 116 additional amino acids compared to ARID1A, which intersperse the ARID1A epitope, and, therefore, the sequence is elongated from 151 to 267 amino acids. These numbers can be visualized when performing a multiple sequence alignment of ARID1A, ARID1B, and ARID2 together using Jalview ([40]; see Appendix A). The antibody epitope region, highlighted with a red box, shows this amino acid stretch in ARID2, as well as a higher sequence similarity between ARID1A and ARID1B in the first 50 amino acids, which is less pronounced towards the end of the epitope. Consequently, we believe there is no interaction of our antibody 182560 with ARID2, and we expect that ARID1B and ARID2 do not affect the quality of our assay.

With this ELISA, we hope to provide a useful tool for research and clinical application in the near future. To use a synthetic lethal effect in tumor cells, knowledge about the actual expression of ARID1A is essential.

## 5. Conclusions

Here, we present an enzyme-linked immunosorbent assay (ELISA) to quantitatively measure ARID1A levels in tumor tissue or cell culture lysates. In brief, lysates are coated on a high affinity plate that has improved conditions for hydrophilic proteins such as ARID1A. To avoid effects due to dilution, total protein concentrations should be above 20 µg/mL. Validation of our assay showed a quantitative range from 0.78 to 50 ng/mL ARID1A and R^2^ of 0.99 regarding the standard curve across all assays performed on different days. The accuracy and precision of the standards meet with 90.26%, and a coefficient of variation with 4.45%, respectively, the EMA and FDA criteria, even across several plates. Additionally, our ELISA clearly distinguished HEK-293T wild-type cell lysates from homozygous *ARID1A* knockout cells. The results were further confirmed by western blot. Furthermore, an analysis of the accrued samples showed that the combination of lysing and measuring with this protocol yielded a precision with a coefficient of variation of 11.78%.

Our ELISA can help to detect and understand protein expression changes in *ARID1A* wild-type and mutated tumors. In the future, it may be used for prognostic and predictive purposes in tumor entities and their therapies, especially since the use of synthetic lethal effects in tumor cells is gaining momentum in phase II studies. Biochemical assays requiring moderate technical equipment and using standard chemicals and consumables like our proposed ELISA method allow a high sample throughput and will accelerate personalized medicine and treatment.

## Figures and Tables

**Figure 1 cancers-15-04096-f001:**
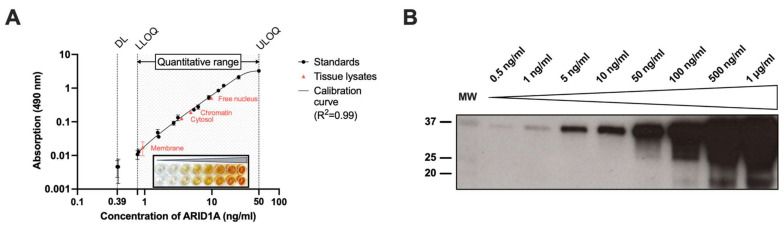
Dynamic range of ARID1A measurements of both ELISA and western blot. (**A**) ARID1A standards and subcellular fractions of tumor tissue lysates were measured in an ELISA assay as described in Materials and Methods as technical replicates (black dots: standards (*n* = 2), red triangles: tumor lysates (*n* = 2)). The absorbance was measured at 490 nm after addition of H_2_SO_4_ (2.5 M) to the colorimetric substrate OPD. A representative standard curve is shown in the black box. DL, detection limit; LLOQ, lower limit of quantification; ULOQ, upper limit of quantification; (**B**) Western blot analysis was performed as described in Materials and Methods, and ab182560 (Abcam, UK) was used as primary antibody (1:2000 diluted in TBS-T). The left lane harbors 6 µL of the molecular weight marker (MW). ARID1A standards were diluted in SDS-PAGE running buffer (25 mM Tris, 192 mM glycine, 1% SDS (*w*/*v*), pH 8.4) to the concentrations annotated above the membrane. Densitometry values obtained with ImageJ [30] are shown in Appendix A.

**Figure 2 cancers-15-04096-f002:**
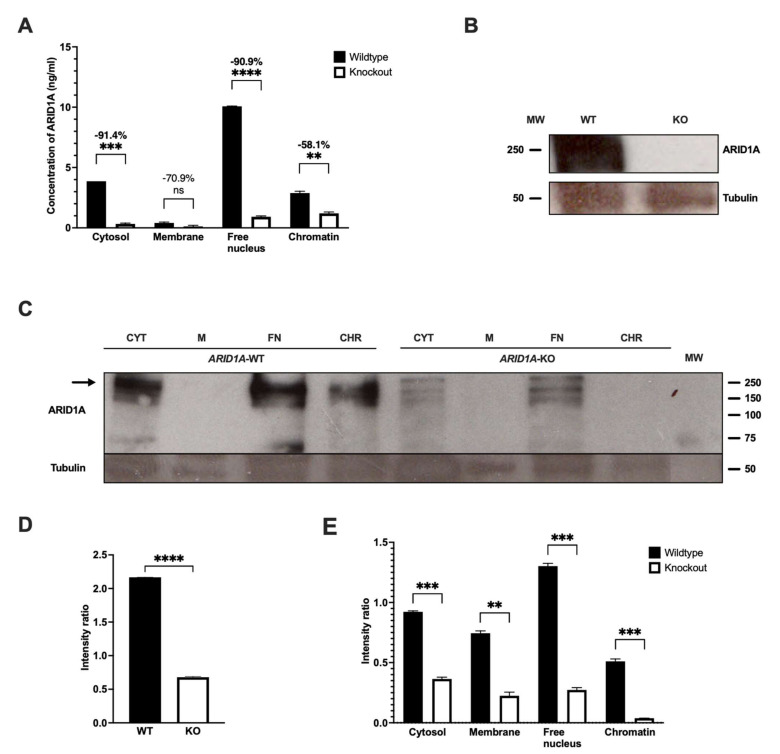
ARID1A levels are significantly reduced in *ARID1A* knockout cells. (**A**) Subcellular fractions generated from HEK-293T *ARID1A* knockout (white) and wild-type (black) cell lines were tested for ARID1A using our established ELISA protocol with the primary antibody ab182560 (Abcam, UK). Student’s *t*-test was performed to determine *p*-values. ns *p* > 0.05; ** *p* ≤ 0.01; *** *p* ≤ 0.001; **** *p* ≤ 0.0001; (**B**) Western blot analysis was performed as described in Materials and Methods, and primary antibody ab182560 was used against ARID1A, while ab7291 was used against alpha-Tubulin. For *ARID1A* knockout (KO) and wild-type (WT) analysis, lyophilized cell pellets ab257250 (Abcam, UK) were used where each lane contains 6 µL of the reconstituted pellet according to the manufacturer’s instructions. The molecular weight markers (MW) at 250 kDa and 50 kDa indicate presumable ARID1A (242 kDa heavy protein) and alpha-Tubulin (50 kDa heavy protein), respectively. (**C**) Western blot analysis was performed as described in Materials and Methods using ab182560 and ab7291 as primary antibodies. HEK-293T cells (ab278824, Abcam, UK) of both *ARID1A* WT and KO were fractionated in cytosol (CYT), membrane (M), free nucleus (FN), and chromatin (CHR). Analysis focused on a band at nearly 242 kDa, in comparison to a molecular weight marker (right lane), which might be specific for ARID1A (arrow); (**D**,**E**) show intensity ratios measured in (**B**,**C**) using ImageJ [30] as well as *p*-values obtained by Student’s *t*-test, respectively.

**Figure 3 cancers-15-04096-f003:**
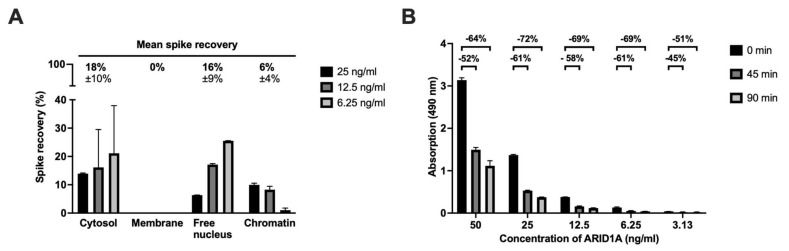
Spike recovery analysis of ARID1A fragment in four subcellular fractions and tests of fragment stability over multiple incubation periods. All samples were measured in duplicate. (**A**) Before performing the standard ELISA protocol, we created three types of samples: (*1: matrix—unspiked*)—*ARID1A* knockout cell pellet fractions (ab278824, Abcam, UK) 1:26 diluted in carbonate buffer; (*2: matrix—spiked*)—*ARID1A* knockout cell pellet fractions of ab278824 1:26 diluted in carbonate buffer spiked with final concentrations of ARID1A fragment of 25 ng/mL (black), 12.5 ng/mL (dark grey), and 6.25 ng/mL (light grey); and (*3: buffer—spiked*) carbonate buffer spiked with final concentrations of ARID1A fragment. Spike recovery, expressed as the proportion of ARID1A fragment re-found in the spiked matrix sample, compared to the spiked pure buffer sample, was then calculated as follows: [(*2*) − (*1*)]/(*3*) × 100. Mean spike recoveries (bold) +/− standard deviations (not bold) at the top of the panel were calculated by averaging all recovery rates at different dilutions within one subcellular fraction; (**B**) Different standard dilutions were incubated on ice for 0 min (black), 45 min (dark grey), and 90 min (light grey) before measuring ARID1A levels. Signal reduction after 45 min is shown in the lower row, while changes after 90 min are presented in the upper row at the top of the panel. Measurements show means and standard deviations of duplicates.

**Figure 4 cancers-15-04096-f004:**
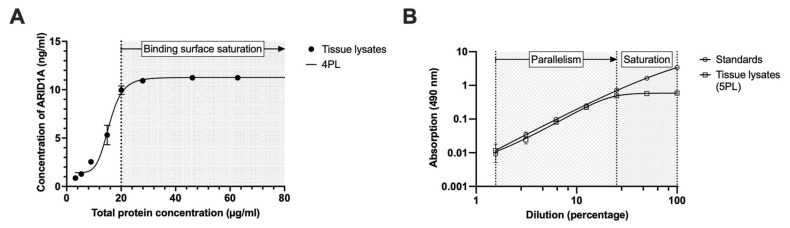
Correlation of ARID1A and total protein concentration and parallelism of lysates and standards. Tumor tissue samples were prepared as described in Materials and Methods. All samples were measured in duplicate. (**A**) Free nucleus fractions and ARID1A-protein standards were coated on one 96-well plate to determine the ARID1A concentration while total protein concentration was determined on another independent plate using a BCA assay with BSA as standard protein. ARID1A concentrations in the lysates were extrapolated based on the ARID1A standard curve. ARID1A concentrations and total protein concentrations were correlated using a 4-parameter regression fit (4PL); (**B**) Crude tumor lysates and ARID1A standards were serially diluted 2-fold for at least six times and coated onto an ELISA plate. After performing incubation steps as described in Materials and Methods for the established ELISA protocol, the absorbance was measured at 490 nm. 100% corresponds to the highest protein concentration measured, which is 62.79 ng/mL. Both curves are interpolated by a 5-parameter regression fit (5PL).

**Table 1 cancers-15-04096-t001:** Materials. All chemicals used were of analytical grade or purchased as concentrated stocks suited for ELISA assays.

Chemicals	Detailed Description
Coating solution	Carbonate (NaHCO_3_/Na_2_CO_3_; ratio: 2:1) buffer (50 mM, pH 9.6)
Washing solution	Phosphate-buffered saline pH 7.4 (PBS) added with 0.1% Tween-20 (=PBS-T 0.1%)
Blocking solution	PBS-T (0.05%) supplemented with 0.2% Casein (I-Block, T2015, Thermo Fisher Scientific, Waltham, MA, USA)
Primary antibody	Primary antibody (ab182560, Abcam, UK) diluted 1:1000 in blocking solution
Secondary antibody	Secondary antibody (anti-rabbit, K4003, Dako, Glostrup, Denmark) diluted 1:2 in blocking solution
Substrate solution	5 mg OPD (o-phenylenediamine dihydrochloride, 34006, Thermo Fisher Scientific, USA) diluted in 9 mL distilled water and 1 mL Stable Peroxide Substrate Buffer (34062, Thermo Fisher Scientific, USA)
Stopping solution	1403 µL H_2_SO_4_ (95–99% purity) diluted in 9 mL distilled water (=2.5 M H_2_SO_4_)

**Table 2 cancers-15-04096-t002:** ELISA protocol. Plates were emptied in between all steps by flicking the plate over a sink and removing the remaining drops by patting the plate on a paper towel. OPD, o-Phenylenediamine dihydrochloride; RT, room temperature.

Steps	Detailed Description	Incubation
Coating	Prepare tumor lysates by diluting the cellular fractions in coating solution 1:26 (total protein concentration should be at least 20 µg/mL). Prepare minimally 6 standards that are serially diluted in the range between 0.78–50 ng/mL. Apply 100 µL of each standard, lysate fraction, and blanks (e.g., carbonate buffer only) at least in duplicate to corresponding wells.	Seal the plate with parafilm. Then, incubate (for 14 to 18 h) overnight at 4 °C.
Washing	Apply 200 µL washing solution. Repeat 2 times.	Incubate at RT for 5 min on a plate shaker.
Blocking	Apply 200 µL blocking solution.	Incubate at RT for 2 h on a plate shaker.
Washing	Apply 200 µL washing solution. Repeat 2 times.	Incubate at RT for 5 min on a plate shaker.
Primary antibody	Apply 100 µL primary antibody diluted 1:1000.	Incubate at RT for 2 h on a plate shaker.
Washing	Apply 200 µL washing solution. Repeat 2 times.	Incubate at RT for 5 min on a plate shaker.
Secondary antibody	Apply 100 µL secondary antibody diluted 1:2.	Incubate at RT for 2 h on a plate shaker.
Washing	Apply 200 µL washing solution. Repeat 2 times.	Incubate at RT for 5 min on a plate shaker.
Application of OPD	Apply 100 µL substrate solution.	Incubate at RT for 30 min in the dark.
Application of H_2_SO_4_	Apply 100 µL stopping solution.	Measure optical density at 490 nm.

**Table 3 cancers-15-04096-t003:** Accuracy and precision of our ELISA assay. All experiments and each plate model were evaluated using the inter-assay accuracy (expressed as percent of the nominal value when performing assays on several days) and the inter-assay precision (expressed as the mean of each analyte’s CV (coefficient of variation). As the subcellular fractions are not represented in the ‘Standards’ column, those fields are termed ‘NA’. Same lysates and different lysates were analyzed in four fractions for which the ‘membrane’ is not shown here due to low signals. In the column ‘Same lysate’, we analyzed one lysate being generated in one session but measured on different days. In the column ’Different lysate’, we evaluated lysates from one tumor being produced in different sessions and diverse regions of the tumor. In both experiments, back-calculated ARID1A levels were then compared by the coefficient of variation in percent (%CV). The column ‘All repeated measurements’ shows the mean %CV of all performed measurements (same lysate and different lysate, *n* = 24) in each fraction. Accuracy calculation was not available (NA) as there is no lysate available with nominated ARID1A concentrations. In the context of lysate analysis, the inter-assay precision reflects the mean coefficient of variation of all fractions.

Deviation in ARID1A Levels	Standards (AE Model ^1^)	Standards (EP Model ^2^)	Same Lysate (*n* = 18)	Different Lysate (*n* = 6)	All Repeated Measurements (*n* = 24)
Cytosol (%CV)	NA	NA	10.05%	12.23%	10.31%
Free nucleus (%CV)	NA	NA	10.76%	26.28%	12.57%
Chromatin (%CV)	NA	NA	11.06%	19.2%	12.46%
Inter-assay accuracy	90.26%	93.16%	NA	NA	NA
Inter-assay precision (%CV)	4.53%	4.45%	10.61%	21.31%	11.78%

^1^ AE model (all experiments model); ^2^ EP model (each plate model).

## Data Availability

There is no data that has been uploaded on any platform.

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
