# Peer review of "Development of an Enzyme-Linked Immunosorbent Assay (ELISA) for the Quantification of ARID1A in Tissue Lysates"

_cancers, 2023, doi:10.3390/cancers15164096_

Round 1

Reviewer 1 Report (New Reviewer)

The authors propose a newly developed ELISA method for measuring ARID1A in cancers. The methodology is good and well explained. I have a couple of points:

-          The authors first stress the importance of a reliable quantitative measure of ARID1A in gynecological cancers, and thus perform ELISA. Why was then WB chosen for validation, given the fact that it is a less quantitative technique?

-          What about detection of mutated vs wt ARID1A? Have the authors taken that into consideration?

Author Response

Reviewer 2 Report (New Reviewer)

I would like to congratulate the authors for a job well done, but I have to ask one important thing. The authors constantly refer to the fact that "there is no easily available method to quantify ARID1A concentrations". However, if you type ELISA and ARID1A into the Internet search engine, you will find many companies that offer such an ELISA kit (e.g. abcam, abbexa). If your kit is different and unique, it is necessary to state this in the introduction or compare the advantages and disadvantages with existing kits.

The graphs should have the same size and also the same font in them for each label.

Please carefully read the description of each graph, e.g. I do not understand the description of Figure 3A

Why is RT * in Table 2. What does the star mean?

Round 2

Reviewer 1 Report (New Reviewer)

n/a

Author Response

Thank you for your work as reviewer.

Reviewer 2 Report (New Reviewer)

Congratulations to the authors for an improved work. However, I still noticed some errors that should be corrected:

In line 110, you used the same phrase twice

In tab.2, put a dot after the sentence in the line for the last washing.

Check in the entire article whether you indicate the correct ARID1A or Arid1a (eg Fig.1)

In the legend of Fig. 2 and in the description, please unite wildtype, wild type or wild-type

Correct the sentence in line 401 so that the word fractions does not appear twice

In line 420 sentence: „All samples were measured in duplicate. (A)“ is mentioned twice

The description of Fig. 3 is a little unclear. Try to at least divide 3 types of samples like 1,2.3

Author Response

Reviewer 2:

Congratulations to the authors for an improved work. However, I still noticed some errors that should be corrected:

- In line 110, you used the same phrase twice.

- In tab.2, put a dot after the sentence in the line for the last washing.

- Check in the entire article whether you indicate the correct ARID1A or Arid1a (eg Fig.1).

- In the legend of Fig. 2 and in the description, please unite wildtype, wild type or wild-type.

- Correct the sentence in line 401 so that the word fractions does not appear twice.

- In line 420 sentence: „All samples were measured in duplicate. (A)“ is mentioned twice.

Answer: Thank you very much for appreciating our additional work we had invested into the previous draft. Thank you for pointing out our remaining mistakes. We deleted all duplicated sentences and adjusted the manuscript where necessary. For example, we consistently used ARID1A (protein) and ARID1A (gene) and checked the entire manuscript for the use of ‘wildtype’, ‘wild-type’ and ‘wild type’. Since ‘wild type’ is a noun and ‘wild-type’ is the corresponding adjective, we hope to have cleared up all irritations.

- The description of Fig. 3 is a little unclear. Try to at least divide 3 types of samples like 1,2.3.

Answer: Thank you for mentioning this ambiguity. We actually had numbers only in a previous version, which seemed to irritate as well, so we added descriptions. We now believe an explanation of the three produced lysates makes it easier to understand for the reader. As a compromise, we added ‘1’, ‘2’, and ‘3’ to the existing explanations, such as ‘matrix-unspiked’.

Upon the remarks of reviewer 2, we re-read our manuscript several times, eliminated several misspellings (e.g., line 68, 100, 267, 366), added punctuation marks (e.g., lines 181, 205, 215), and removed or added articles (e.g. line 299), where appropriate and where we unfortunately overlooked them in the previous versions. All changes are, of course, visible via track changes and do not at all alter the content or understanding of our manuscript.

This manuscript is a resubmission of an earlier submission. The following is a list of the peer review reports and author responses from that submission.

Round 1

Reviewer 1 Report

In the work of Hinsberger and co-workers, authors have described the development of the ELISA assay to determine the concentration of AIRD1A protein. As this protein has a prognostic value for a cancer patient, this investigation might be beneficial and of interest to clinical practice.

Before further considering for publication, the authors should add the experiment in which they will add various concentrations of standard AIRD1A to the lysate of ARID1A -knockout cells. It is possible that the concentration of the protein bound to the ELISA plates depends on the "matrix" and that its interaction with the ELISA plate differs when other proteins and other biomolecules are present. Therefore, to ensure that the ARID1A concentration in cancer cell lysate is objectively determined, the standard curve should be prepared by performing the indirect ELISA on the cell lysate spiked with varying concentrations of the ARID1A protein.

In addition, the method should be compared with other, well-established, and previously performed assays, for instance, with WB has done with varying concentrations of ARID1A.

Furthermore, the manuscript text has numerous unnecessary repetitions, which make it much longer, but not informative. For instance, the methods section should contain a detailed description of methods, not the literature background. This helps explain unexpected results (Fig.2) obtained with the indirect ELISA.

The discussion contains repetitions from the Introduction, and this must be rewritten. The authors must focus on the method's results, drawbacks, and advantages.